# Conservation Analysis and Colorimetric Characterization of Betalain Extracts from the Peel of Red Beetroot, Golden Beetroot, and Prickly Pear Applied to Cottage Cheese

**DOI:** 10.3390/foods14020228

**Published:** 2025-01-13

**Authors:** Elizabeth López-Solórzano, Claudia Muro, Yolanda Alvarado Perez, Andrea Y. Guadarrama-Lezama, Elsa Gutiérrez-Cortez, Juan Manuel Urrieta

**Affiliations:** 1División de Estudios de Posgrado e Investigación, Instituto Tecnológico de Toluca, Tecnológico Nacional de México, Avenida Tecnológico S/N, Colonia Agrícola Bellavista, Metepec 52149, Estado de México, Mexico; dd22280272@toluca.tecnm.mx (E.L.-S.); yalvaradop@toluca.tecnm.mx (Y.A.P.); 2Facultad de Química, Universidad Autónoma del Estado de México, Paseo Colón esq. Paseo Tollocan s/n, Col. Residencial Colón, Toluca 50120, Estado de México, Mexico; 3Facultad de Estudios Superiores Cuautitlán, Universidad Nacional Autónoma de México, Km 2.5 Carretera Cuautitlán-Teoloyucan, San Sebastián Xhala, Cuautitlán Izcalli 54714, Estado de México, Mexico; elsaneqpm123@gmail.com; 4División de Estudios de Posgrado e Investigación, Instituto Tecnológico de Villahermosa, Tecnológico Nacional de México, Km. 3.5 Carretera, Villahermosa—Frontera, Cd Industrial, Villahermosa 86010, Tabasco, Mexico; juan.us@villahermosa.tecnm.mx

**Keywords:** betalains application, red beetroot, golden beetroot, purple prickly pear, stability, cottage cheese

## Abstract

The maintenance of betalains and the color of extracts from the peel of red beetroot (RBAC), golden beetroot (YBAC), and purple prickly pear (PBAC) were evaluated, describing the capacity of their use as natural pigments and in the formulation of attractive and functional foods. Betalain extracts were prepared as juices from frozen and dehydrated peel, adding organic acids and concentrating for water reduction. Extracts were evaluated and applied on cottage cheese, measuring the capacity of betalains retention and pigmentation, during 10 days of storage of closed and opened products. Extracts of RBAC showed the highest betacyanin concentration, followed by YBAC with betaxanthins and PBAC with less betacyanin content. The pH stability for the extracts was pH4–7; RBAC and PBAC were stables at <90 °C, whereas YBAC exposed >125 °C. Extracts were constant during 10 days under oxygen and light exposure; however, YBAC exhibited low resistance in this environment. With cottage cheese, extracts exposed no changes in betalains and color on closed products (10 days of storage at 4 °C). In opened products, PBAC maintained the maximum betalains and color at 90%, PBAC at 75%, and YBAC at 60%. The obtained data contributed to use of agro-industrial residues, betalain extraction and conservation, and their potential use in food coloration and stabilization.

## 1. Introduction

Currently, the use of natural pigments in food areas is becoming more common because they offer color and nutrient value for replacing synthetic colorants, which have been shown to have serious effects on human health, such as cirrhosis, genotoxicity, neurotoxicity, and deficits in learning and memory [1,2].

Natural pigments feature betalains; they are hydrophilic biomolecules containing betacyanin’s (red-violet color) and betaxanthins (yellow color) [3]. Additionally, betalins also are antioxidants; consequently, they are considered potential colorants, enhancing the appearance of food and maintaining the uniformity of products [4,5].

The major sources of betalains are red beetroots, golden beetroots, and prickly pear. The betalains are extracted from the peel and pulp of these fruits and vegetables. Therefore, the agri-food residues of these materials also contain a great amount of betalains [6,7].

Currently, betalains are produced and commercialized as food additive pigments (E-162), and they are obtained in powered form from dried sources, extracts, and extracts encapsulating spray-dried powders. However, betalains have restrictions in terms of of application because they are extremely sensitive to pH, temperature, oxygen, light, water activity, and enzymatic action, limiting their use as food additive pigments [8].

The identifiable constraints of betalains are their short shelf-life, changes in coloration during food storage, flavor modification, and the total loss of color. In consequence, betalain stability constitutes an important topic of study in the area of natural food pigments, generating data from techniques and methods of extraction, stabilization, and conservation to overcome the mentioned restrictions. For example, Hazervazifeh et al. [9] reported the best conditions of betalain extraction from red beetroot by a microwave technique. Otálora et al. [10] extracted betalains from red beetroot by blanching–cutting–drying of beetroot to analyze the thermal stability of betalains. Boravelli et al. [11] extracted betalains from crude extract, using Polyethylene Glycol and ammonium sulfate for the sugar’s removal in liquid–liquid extraction. Morales-Huerta et al. [12] encapsulated betalains for their stabilization. The authors utilized dried beetroot extract, using foam by egg albumin.

Oktay et al. [13] tested ultrasound-assisted extraction (UAE) to achieve high betalain content from prickly pear fruit (*Opuntia Ficus-indica* L.). Ferreira et al. [14] obtained a betalain extract from *Opuntia Ficus-indica* peel by hydrothermal extraction; after, the extract was encapsulated by spray-drying to produce powder.

In turn, the current topic is betalain application for food pigmentation [15,16], involving different studies to evaluate their capacity of coloration and maintenance in food products. However, more data are necessary to establish the viability and variability of betalain use. Yogurt and milk are the most studied matrices of betalain application. Ghasempour et al. [17] tested different formulations of betalains from red beetroot and basil seed gum to enhance the properties of a probiotic yogurt. In turn, Schneider-Teixeira et al. [18] registered the conservation of betalain extracts from Alternanthera brasiliana on yogurt. Güneşer [19] registered the stability of betalain pigments from beetroots in thermal treatments of cow milk. In turn, Tamer et al. [20] reported the color preservation of betalain extract from Amarant and beetroot on ice-cream formulations.

On other food matrices, Chaari et al. [21] tested betalain extracts from *Opuntia stricta* peel on raw refrigerated beef meat and compared it with Allura red E129. The betalains showed different improvements in regard to the color of meat, decreased the chemical oxidation, enhanced the sensory characteristics, and inhibited the L. monocytogenes replication and transcription process. Mehta et al. [22] reported the application of encapsulated betalain extracts from *Opuntia Ficus-indica* in the preparation of gummies, indicating acceptable behavior in the matrix.

Based on the above reports, the investigation of the betalains stability and their application in food products should be focused to increase the knowledge on this topic and provide tools to evaluate their behavior in food use [1,2,15,16].

The aim of this research was obtaining betalain extracts from agro-industrial wastes of red and golden beetroot and purple prickly pear (*Opuntia lagunae*) and comparing the pigment capacity and stability on cottage cheese to produce attractive and functional foods with effective coloration. The study looks at the extract preparation, a shelf-life study of betalains, the colorimetric stability of betalains, and the extracts’ applications in cottage cheese, analyzing the outstanding food with coloration, the shelf-life of betalains and the shelf-life of cottage cheese. The obtained data involve the contribution of natural pigments and their application in the food area to obtain commercial products.

## 2. Materials and Methods

### 2.1. Materials and Reagents

Whole agro-industrial residues of the peel of red beetroot (*Beta vulgaris* L.), yellow beetroot (*Beta vulgaris* L.), and purple prickly pear (*Opuntia lagunae*) were provided by a food company in the State of Mexico for their use as betalain sources.

Ascorbic acid and acetic acid were bought by J.T. Baker (México). All chemical materials used for the characterization and analysis were of analytical grade. Deionized water was used in all the experiments.

### 2.2. Betalain Extracts Preparation and Conservation Treatments

Samples from the peel of red beetroot, golden beetroot, and purple prickly pear were stored in the individual samples (250 g) and frozen at <−20 °C in hermetic plastic bags.

Betalain extracts were obtained according to following steps. (1) Betalains extracts were obtained from previous frozen samples, which were later defrosted, and the pressing of sample sources, using a microwave for 20 min. The procedure was repeated until the juice from the source was exhausted, recollecting the extracts from each procedure. (2) Betalain extract refining. Obtained extracts from beetroot were centrifuged at 3500× *g* (10,000 rpm) for 5 min. The supernatant liquid was passed through filter paper (Whatman 1). In turn, extracts from the prickly pear were previously filtered by open mesh cotton cloth to remove large solid waste and fiber.

### 2.3. Betalain Extracts Conservation

The treatment of betalain extracts conservation consisted of two steps. (1) The addition of one organic acid, which was selected previously for its use as a conservation agent. (2) The concentration of extracts by water evaporation.

Ascorbic acid was added to betalain extracts from red beetroot and purple prickly pear, whereas acetic acid was added in the betalain extracts from golden beetroot (1% *w*/*v* or 1% *v*/*v*, respectively). To dissolve and homogenize the extracts with the acid, samples were subjected to an ultrasound bath (Bransonic CPXH1800, Emerson, Ferguson, MO, US) at 25 °C for 15 min.

The concentration of extracts was carried out in a rotary vacuum evaporator (IKA RV10 basic, Cole-Parmer, Wertheim, Germany) at 35 °C to achieve 70–80% of water removal. The final betalain extract samples (1 L) from each source were obtained and distributed in amber flasks of 100 mL and stored at 4 °C in the absence of light for later analysis. For later analysis and use, membranes of 0.45 µm pore size and a nylon filter were used to obtain betalain upgrade extracts.

The obtained betalain extracts were identified as extracts with acid and were concentrated from red beetroot (RBAC), golden beetroot (YBAC), and purple prickly pear (PBAC).

For control samples of the betalain extracts of RBAC, YBAC, and PBAC, we included samples of extracts without acid, being identified as RB for red beet, YB for golden beet, and PB for purple prickly pear; subsequently, we utilized extracts with acid (RBA, YBA, and PBA) and extracts without acid which were concentrated (RBC, YBC, and PBC).

### 2.4. Physicochemical Characteristics of the Betalain Extracts and Extraction Yield

In regard to the betalain extracts indicated in the method, they were characterized by extraction yield (mL/g), colorimetric characteristics and physicochemical properties, including betalain concentration, Brix grades (°Bx), pH, and total dissolved solids (TDS).

(1)The extraction yield was determined by Equation (1), where V = volume of extract in (mL) and mi = initial sample weight in (g).


(1)
$$Yield\left[\frac{mL}{g}\right]=\frac{V}{mi}$$


(2)Betalain quantification was obtained following the method described by Oktay et al. [13] using Equation (2).

(2)$$BC\left[\frac{mg}{L}\right]=A*DF*MW*\frac{1000}{E*l}$$
where BC = betacyanin or betaxanthin content in the extracts; A = absorbance λmax 538 nm for betacyanins and 480 nm for betaxanthins; DF= dilution factor (final dilution volume/volume of aliquot to be diluted); MW = molecular weight 550 g/mol for betacyanin and 308 g/mol for betaxanthins; E = molar extinction coefficient 60,000 L/(mol·cm) in H_2_O for betacyanin and 48,000 L/(mol*cm) in H_2_O for betaxanthins; and l = cell length in cm.

The reading at 600 nm was used to correct the absorbance of the impurities. The results were expressed in mg of betacyanin (betanin)/g or betanin/L and mg betaxanthin (vulgaxanthin)/g or mg/L. The total betalain content was calculated as the sum of both molecules.

(3)Determination of the total dissolved solids was carried out using the weight loss drying technique following Equation (3).

(3)$$TDS[\frac{mg}{L}]=\frac{\left(P_{2}-P_{1}\right)*1000}{Vo}$$
where P_1_ = weight of empty capsule (mg); P_2_ = weight of capsule with dry sample (mg); V_0_ = initial sample volume (mL).

(4)The Brix of the samples was estimated using a portable digital Brix refractometer (OPTI model, Bellingham + Stanley, Barcelona, Spain). A volume of 2 mL of sample at room temperature (20 °C) was placed in the test reader. The Brix value was read directly from the instrument. The refractometer was calibrated with distilled water before measuring.(5)Colorimetric analysis was carried by a colorimeter (VINCKOLOR Pro, Hangzhou, China), using the coordinates L* (luminosity), a* (coordinates red/green) and b*(coordinates yellow/blue) of the CIELAB color system.

By colorimetric coordinates, Hue color (H) was evaluated by Equation (4) and Chroma* (chromatic intensity or purity of color) by Equation (5). L*, H, and Chroma* were obtained for each betalain extract.(4)$$Hue=arctan(\frac{b*}{a*})\times 180/\pi$$(5)$$Chroma*={\left({a*}^{2}+{b*}^{2}\right)}^{1/2}$$

### 2.5. Shelf-Life Analysis of Betalain Extracts

The shelf-life analysis was carried out to evaluate the effect of the maintenance method on betalain extracts. The shelf-life was found by measuring the concentration of total betalains in extracts at initial time, t = 0 days until t = 180 days of stored samples as the final condition, with the data being measured every 30 days for 6 months. Betalain quantification was obtained following the method described by Equation (2).

The change in coloration extracts was measured by colorimetric analysis using the color difference by Equation (6) based on colorimetric parameters.

L*_0_, a*_0_, b*_0_ represent initial conditions; L*, a*, b* represent the final conditions [23].(6)$$\mathrm{Color}\;\mathrm{difference}=\Delta E=\sqrt{\left({\left({L*}_{0}-L*\right)}^{2}+({{a*}_{0}-a*)}^{2}+({b*}_{0}-b*{)}^{2}\right)}$$

Samples of the test involved RBAC, YBAC and PBAC, including the sample control, RB, RBA, RBC, YB, YBA, YBC, PB, PBA, and PBC.

### 2.6. Color Stability Analysis of Betalain Extracts

The color stability analysis of the betalain extracts involves integrated thermal stability, pH changes, and oxygen attacks. The analysis was obtained through the total content of betalains and the colorimetric evaluation of betalain extracts due to changes at different pH levels, temperatures, and days of exposure to oxygen.

The betalain content in extracts was assessed according to Equation (2), whereas color properties and the stability of extracts was measured by the colorimetric properties L*, a*, b*, Hue, and Chroma* by Equations (4) and (5) [23].

### 2.7. Application of Extracts in Cottage Cheese

Betalain extracts were applied to commercial cottage cheese, using 0.5% of the weight of the extract.

In regard to the storage times of cottage cheese, the coloration changes were evaluated by a colorimetric technique, in accordance with Section 2.3 of the manuscript, using the color parameters Hue, Chroma, and color difference by Equation (6).

The measuring was performed directly on the cheese surface 10 min after opening the recipient of storage, selecting three random readings at separate locations per sample. For the sample control, we used close single cottage cheese, a close single extract, and a close cheese sample with betalain extract.

### 2.8. Statistical Analysis

Statistical analysis was carried out on betalain extracts through the application of extracts to cottage cheese. The analyses were carried out in triplicate and the results were given as the mean value with standard deviation. The statistical difference was evaluated by an analysis of variance, (ANOVA) followed by a Tukey test using Minitab software version 19.2020.1.0 (Minitab, LLC Software, Inc., State College, PA, USA). Differences were considered significant at *p* < 0.05.

## 3. Results and Discussion

### 3.1. Physicochemical Characteristics and Yield of Betalain Extracts

Table 1 exposes yield, physicochemical characteristics and betalain content in extracts of betalain from red beetroot, golden beetroot, and red beetroot (RBAC, YBAC, and PBAC), including the extract controls as single extracts (RB, YB, and PB), extracts with acid (RBA, YBA, PBA), and extracts without acid and which are concentrated (RBC, YBC, YBC, and PBC).

The betalain extracts of RBAC, YBAC, and PBAC showed different information, indicating significative differences in betalain content and other parameters (*p* < 0.05).

(1)The yield of betalain extracts was obtained with the following order: YBAC > PBAC > RBAC. The difference in yield was associated with the origin (red beetroot, golden beetroot, and purple prickly pear) and water content of each source, exposing the final water content (after the concentration process) of 20% PBAC, 15% YBAC, and 12% RBAC.(2)As expected, the betalain extracts for RBAC, YBAC, and PBAC showed an acidic pH because they contain organic acid.(3)In turn, the RBAC, YBAC, and PBAC exposed different ranges of °Bx and TDS. The highest values of °Bx and TDS were found in RBAC and YBAC because beetroot contains more sugar than prickly pear.(4)Concerning betalain content, the data between the extracts of RBAC, YBAC, and PBAC also showed significative differences (*p* < 0.05).

Betalain extract control showed that the conservation treatments positively affected the betalain content, exhibiting the highest values in RBAC, YBAC, and PBAC. However, RBAC showed the highest value; subsequently, YBAC and PBAC showed the highest values, indicating significative differences between betalain sources (*p* < 0.05).

According to conservation treatments, the control samples indicated distinct affectations. The control extracts for RB YB PB (without any treatment) registered the lowest betalain contents.

Extracts treated with ascorbic acid (RBA and PBA) showed similar behaviors to RB and PB, but with an increase in the betalain content, while the extract treated with acetic acid showed an increase in the betalain content (betaxanthins) in YBA.

The increase in betalains by acid treatment and concentration was associated with the effect of betalain extraction, stabilization, and conservation by the glycosylation process.

Betalain maintenance by the addition of ascorbic acid was also reported by Gandía-Herrero et al. [24] and Wang et al. [25]. These authors showed that ascorbic acid slightly changed the pH of the betalain extraction medium, causing the formation of quinones by the effect of polyphenol oxidases, which stabilized the betacyanins.

Treated extracts by sample concentration, due to water evaporation without acid (RBC, YBC, and PBC), registered complete or partial degradation. Betacyanins and betaxanthins were not detected in the Uv-vis spectra. The degradation of betalains in the control extracts of RBC, YBC, and PBC was linked to the absence of organic acid as well as evaporation temperature, oxygen, light, and the high sensibility of betalains from beetroot.

The high betalain content in RBAC was registered as betacyanin because betaxanthins were not detected by this method. However, the control samples for RB and RBA showed the survival of betaxanthins, whereas RBC (concentrated extracts) did not register betaxanthins.

In turn, extracts of YBAC observed single betaxanthin content with a betacyanin absence due to the yellow color. The above assumption was confirmed with the control samples YB and YBA because they showed similar colors and single betaxanthins in the Uv-vis spectra.

Subsequently, the PBAC extracts showed single betacyanin content. Similarly, the control extracts showed similar behavior; however, the wide band of betacyanins also suggested the presence of betaxanthins.

Figure 1 shows the Uv-vis spectra of the betalain extracts of RBAC, YBAC, and PBAC and the control extracts of RB, YB, and PB, confirming the above data.

Uv-vis spectra from the RBAC, YBAC, and PBAC and the control extracts of RB, YB, and PB displayed betalamic acid with conjugated dienes of the 1,7-diazaheptamethine and the chiral center at C-6, exhibiting bands in both regions (Uv and vis) [26].

Bands in the vis range, exhibited betacyanin (420–540 nm), providing violet–red coloration, whereas betaxanthins (320–480 nm) showed a brilliant yellow coloration [25].

The extract of YBAC proved to be single band, indicating the predominance of betaxanthins, whereas the extracts of RBAC and PBAC exposed similar Uv-vis spectra, exhibiting band ranges of 420–580 nm of a betacyanin majority. However, RB revealed the presence of betaxanthins and betacyanins. Therefore, betaxanthins could be hidden in betacyanins due to the concentration of extracts, and the extensive band of betacyanins could obscure the betaxanthins, occurring at 480 nm. The extracts of PB not showed betaxanthins; however, Wang et al. [27] showed that prickly pear contains betaxanthins. Therefore, they could be covered by the betacyanin band.

The apparent betacyanin predominance in the RBAC and PBAC Uv-vis spectra is due to optical activity because they contain two chiral carbons (C2 and C15). In addition, they also contain a hydroxyl group linked to the C6 carbon and a glucosyl group on the C5.

In turn, the glycosylation and acylation of betacyanin provide 5-O- or 6-O-glucosides, resulting in predominant betacyanin structures, including betanin (betanidin 5-O-ß-glucoside).

According to previous studies, betalain bands from beetroot, golden beetroot, and prickly pear might include betacyanin as betanidin (541 nm), betanin and isobetanin (537 nm), 2-descarboxy-betanin (532 nm), 6-O-Malonyl, 2 descarboxy-betanin (535 nm), prebetanin (538 nm), and Neobetanin (267, 306, 470 nm), while betaxanthins could describe the occurrence of indicaxanthin (260, 305, 485), betaxanthin and vulgaxanthin I, II, III, and IV (469–470 nm), and valine–betaxanthin [25,28,29,30].

In comparison with previous studies, there are several reports revealing information on betalains; however, the difference in methods of extraction and extraction sources causes diverse data [2,4]. For instance, Ferreira et al. [14] obtained a maximum yield (47.9%) of betalain extraction from the peel of Rose prickly pear. However, the authors used high water ratios for solvent extraction, equivalent to the volume of the betalain extracts. In addition, the authors did not mention the calculi base to express the data in a percentage %.

In regard to betalain content, the data from RBAC agreed with the data reported by [19]. Authors detected betanin content of up 8222 mg/g of fresh weight [FW] from beetroot, while golden beetroot registered the lowest betaxanthin content at 193.7 mg/g vulgaxanthin I. However, the authors obtained betalain extracts by solid–liquid extraction, with methanol and ascorbic acid acting as the extraction solvent.

Other studies of red beetroot reported the lowest betalain content; Prieto-Santiago et al. [24] studied three beetroot samples, including thermally treated juice, puree, and a whole beetroot. As a solvent, they used a ethanol–water solution (50:50 *v*/*v*) that was shaken for 15 min. Juices and puree showed the highest betalain content (835 and 868 mg/kg, respectively). Silva et al. [31] studied extracts from powered red beetroot (previously dread at 50 °C) using ultrasound extraction (UAE) and 75 mL/g of water as solvent at 30 °C and 30 min. The extracts registered 7 mg/g of betalains, including 4.45 mg/g of betacyanin and 2.42 mg/g of betaxanthin.

In turn, Masithoh et al. [32] detected the lowest betalain range, indicating 0.17–0.30 mg/g in extracts from dried beetroot samples (70 °C). However, the authors used methanol as an extraction solvent and stirred it at 180 rpm for 30 min at room temperature. Chhikara et al. [33] used dried red beetroot (55 °C) to obtain aqueous extracts of betalains by solid–liquid extraction, registering 62 mg/L betacyanin’s and 61 mg/L betaxanthin, whereas Attia et al. [34] used a solid–liquid extraction with acidified ethanol 2%, using citric acid and a blender for 15 min at room temperature. The extracts were concentrated under a rotary vacuum evaporator at 40 °C, registering 3.8 mg/g of betalains per FW.

Similarly, the reports from purple prickly pear indicate distinct information on betalain content. Mehta et al. [22] reported a high content (858.28 mg/L) of betalains in extracts from the mixture of the pulp and peel of prickly pear (*Opuntia ficus-indica*) using glycerol for solvent extraction and UAE (30–60 °C) (10–30 min).

In turn, Oktay et al. [13] revealed low betalain content (0.472 mg/g) in extracts from dried prickly pear using UAE (30 min extraction time, 49.99 °C, 40% ethanol concentration, and a 1/30 solid/solvent ratio). Chaari et al. [21] studied betalain extracts from dried peel of Rossa prickly pear, using sequential extraction by water at room temperature in light-protected glass flasks during 24 h of stirring at 200 rpm. Betalains were added to the extracts to enhance stability; however, they registered a low betalain range (14.8–41.2 mg/g).

With golden beetroot, further data were not found in the literature because this source is not extensively known in the pigments area.

### 3.2. Colorimetric Characteristics of Betalain Extracts

Figure 2 shows coloration characteristics (Hue and Chroma*) of the fresh betalain extracts of RBAC, YBAC, and PBAC (pH4, temperature 4 °C), as well as their control samples at equal conditions.

According to the origin source and extract control, the betalain extracts showed the distinct color characteristics Hue and Chroma*, which were linked to a betalain majority and color degradation [33].

The extracts of RB and RBA exhibited a red Hue, due to betacyanin (betanin) and betaxanthin (vulgaxanthin) predominance, whereas RBAC had a violet Hue, indicating a high betanin and betaxanthin ensemble. RBC presented a brown Hue, showing color degradation and betacyanin and betaxanthin degradation. In turn, the Croma* was highest in RB and RBA in comparison to RBAC because the red color has a superior purity than the violet color.

In the case of the extracts of YBAC, YB, and YBA, they have a high betaxanthin content, showing a yellow color, and a high Croma*, suggesting a high color purity, which was increased by the addition of acetic acid and the concentration procedure. However, YBC did not retain the betalains, showing a degradation in color.

Although, the extracts of RBAC and PBAC presented betacyanin predominance, both showed a different Hue. PBAC exposed a red Hue, showing a betanin predominance with the highest Croma*; extracts PB and PBA showed violet coloration and thus the lowest Chroma*. PBC did not expose color degradation. Therefore, the extracts of PBC and PBAC present disperse betalains by the degradation of betaxanthins, betalamic acid, and 2-hydroxy-2-hydrobetalamic acid from beetroot, which is superior in betaxanthins.

### 3.3. Shelf-Life Analysis of Betalain Extracts and the Effect of Organic Acid Addition and Concentration

Table 2 shows data regarding betalain stability and the storage time versus betalain content in extracts RBAC, YBAC, and PBAC, including the sample control extracts. Data involve the initial betalain content and 180 days of the extract’s storage (4 °C and amber flasks).

In addition, Figure 3 shows the kinetic behavior of betalain extracts, complementing the information of Table 2.

Overall, the extracts of RBAC, YBAC and PBAC revealed high betalain maintenance percentages during the 180 days of storage due to treatment with acid and concentration. However, between extracts, they observed different behavior in terms of maintenance (*p* < 0.05).

The highest betalain conservation was observed in YBAC because they indicated a 95% betalain maintenance. Subsequently, RBAC showed 87% of betalains during 180 days of storage, whereas PBAC exhibited 50%.

According to kinetic of betalains, after 360 days, the extracts of YBAC exhibited the highest betalains retention (70%), while RBAC had 30% and PBAC featured the lowest betalain retention (10%).

The high shelf-life of the extracts of PBAC and YBAC indicated the positive effect of ascorbic acid or acetic acid addition and extract concentration. However, the conservation method was not sufficient in regard to retaining the betalains from PBAC, as the water content in this extract was superior to RBAC.

In turn, the control sample showed a short shelf-life, registering 100% of betalain degradation after 1–5 h of extraction and storage, evidencing the positive effect of the conservation method. The coloration change (discoloration) in betalain extracts from control extracts RB, YB and PB were linked to the activity of endogenous enzyme ß-glucosidases on betaxanthins, releasing cyclo-DOPA-5-O-ßglucoside.

Current investigations of Mehta et al. [22] showed comparative data on the shelf-life of betalains. Authors observed 78.65 % of the color retention of betalains during 70 days of storage under amber and 4 °C conditions. However, the predictive kinetic indicated 404 days of betalain maintenance. The shelf-life was measured in samples from encapsulated and lyophilized extracts from prickly pear Ficus-indica. The storage temperature was the principal factor affecting the shelf-life of betalains, whereas the high stability was due to use of glycerol as an extraction solvent; however, the encapsulation also influenced these results.

### 3.4. Color Stability of Betalain Extracts

Table 3 and Table 4 show data regarding betalain retention and color parameters, exhibiting the stability of the extracts of RBAC, YBAC, and PBAC under different conditions of exposure in the pH range of 4–9, the temperature range 20–125 °C for 30 min, and oxygen and light for 20 days with agitation at 150 rpm. The referent conditions of the study were the extracts of RBAC, YBAC, and PBAC at pH4, temperature = 4 °C, and null exposition to light and oxygen.

In general, extracts exhibited changes in the betalain content, betalain retention and color difference ΔE by effect of exposure conditions.

According to the pH, RBAC showed the highest amount of stable betalains, with the pH ranging 4–7; however, at pH > 7, RBAC manifested betalain loss with 50–18% of betalain retention. In consequence, the color difference at pH < 7 was ranging ΔE = 7 and pH > 7, ΔE = 29, indicating high betalain degradation and color at pH 8–9.

In turn, betalains from RBAC were thermically resistant to up to 90 °C, manifesting the color difference ΔE = 33. However, the drastic thermal degradation of betalains was observed at the temperature >100 °C, manifesting 30% of betalain retention and a brown color with ΔE = 74. The drastic changes in color in the extract of RBAC by the temperature increasing was observed at >100 °C, modifying a red color to brown.

Concerning oxygen and light exposure, betalains from RBAC showed high resistance. In this case, betalains manifested a betalain permanency >10 days of light and oxygen exposure with ΔE = 25; however, at 20 days of exposure, the brown color was predominant in the extracts. Nevertheless, it is an important result because, by organic acid and th concentration of extracts, RBAC resisted the oxygen and light attack.

Similarly, the extracts of PBAC exposed a betalain stability at pH < 7; however, at pH > 7, the betalains were reduced, exhibiting a brown color due to degradation of betalains.

In comparison with RBAC, the extracts of PBAC exposed low betalain resistance to pH. In addition, the betalains were thermically stable at up <90 °C, showing a low stability at >90 °C. According to Herbach et al. [35], the betalains from prickly pear are stable until <80 °C, prompting the isomerization of indicaxanthins. However, herein, the yellow color at 100 °C was observed. Consequently, the degradation of betaxanthins was suggested at >100 °C.

By exposure to oxygen and light, betalains from PBAC showed high resistance, indicating low ΔE in comparison to extracts from RBAC. The color can be observed in Table 4 at 20 days of oxygen and light exposure.

In turn, the extracts of YBAC were not affected by pH; they maintained the betalains in the pH range of 4–9. In addition, the extracts of YBAC was thematically resistant because, in the temperature range 20–125 °C, they did not show any lost betalains. However, the oxygenation and light affected the betalains, showing 50–38% of betalain retention and changes from yellow coloration to green coloration.

In resume, the stability data from RBAC, YBAC, and PBAC showed that betanin from RBAC and PBAC have great stability at the pH range 4–7, favoring the recondensation of betalamic acid or cyclo-DOPA-5-O-ß-glucoside; however, under alkaline conditions, the hydrolysis of the aldimine bond was predominant, causing betalain instability. In addition, alkaline pH influenced the decomposition of betanidine into 5,6-dihydroxyindole-2-carboxylic acid and methylpyridine-2,6-dicarboxylic acid, affecting the betalains’ composition and coloration. In turn, the betaxanthins were stable at a pH range of 4–7; however, the oxidation of vulgaxanthin I in RBAC and PBAC occurred outside these conditions because alkaline pH causes the degradation of indicaxanthin, generating protocatechuic, which blocks the regeneration of betaxanthins. However, it does not occur in YBAC, showing high stability in regard to acid and alkaline pH.

Relating to thermal resistance, betacyanins from RBAC and PBAC were stable at the range < 90 °C (at acid pH). The thermal changes occurred at ranges >90 °C, achieving a yellow to brown color with the degradation of betanin to neobetanin, dehydrogenation and decarboxylation of neobetanin, and the polymerization of betalains, releasing betalamic acid and colorless cyclo-DOPA-5-O-glycoside [36]. However, high stability was observed in betaxanthins from YBAC.

In turn, betalains from RBAC, PBAC, and YBAC were resistant to oxygenation and light, achieving stability during 10 days of oxygen and light exposure.

The stability data from RBAC, YBAC, and PBAC were partially coincident with the recollected information of previous studies [36] because the extraction methods are distinct and there is different information between stabilization analysis and the environment of exposure.

Lombardini et al. (2021) [37] showed that the best stability range of enzymatic betalain extracts from beetroot were 4–25 °C, highlighting the high betaxanthin stability in terms of the constant degradation rate. Under the mentioned conditions, the UV-light did not affect the pigment coloration; however, ranges > 40 °C altered the pigmentation.

Otalora et al. [36] reported a study of betalain extracts involving a beetroot blanching treatment (immersion in water at 90 °C for 7 min). After, they lyophilized to produce powered betalains and thermically analyzed the products. The author found low thermic resistance in the betalain extracts because the treatment at 90 °C affected the betalain content, whereas that the betalins powder present highest thermal resistance.

In turn, Mehta et al. [22] evaluated the thermal stability (80–180 °C, 30–60 min) of the encapsulated color pigment of betalains from prickly pear. They reported the highest thermostability of encapsulant material; however, the color difference ΔE incremented with the temperature increment, achieving a brown color.

### 3.5. Shelf-Life Analysis of Betalain Extracts on Cottage Cheese

Table 5 shows the shelf-life of betalains from the extracts of RBAC, YBAC, and PBAC on cottage cheese (CCH), showing the betalain content and color permanency at storage times t = 0 and t = 10 days with the refrigerated conditions T = 4 °C and once the cheese was opened.

Samples CCH + RBAC, CCH + YBAC and CCH + PBAC represent samples of cheese with 5% W of betalain extracts. The control samples were CCH without extracts, as well as the extracts of RBAC, YBAC and PBAC for closed product and opened product. Opened products refers to opened products used for frequent consumption that are exposed to oxygen and light.

After 10 days of storage at 4 °C, closed products CCH + RBAC, CCH + YBAC and CCH + PBAC did not observed changes in betalains and color. However, products that are frequently opened for consumption showed the following changes, indicating significative differences between closed and opened products (*p* > 0.05).

CCH + RBAC registered 92% in regard to betalain maintenance with red color at initial and final period of storage.

CCH + YBAC exhibited yellow coloration; however, after 10 days of storage, samples observed slight changes in the yellow color, exposing 60% of betalain retention.

In turn, CCH + PBAC showed 85% in terms of betalain retention, exhibiting a violet color at the initial and final storage times.

The data were coincident with the above study of betalain stability, showing that the extracts of PBAC resulted with highest capacity of survival under stress conditions of oxygen and light, followed by RBAC, whereas the extracts of YBAC had a limited resistance under oxygen and light conditions. However, the results of the application of betalain extracts were excellent in regard to providing coloration in cottage cheese.

Matching the study of betalain extract application, Figure 4 exhibits kinetics of the color characteristics L* and ΔE from samples CCH + RBAC, CCH + YBAC and CCH + PBAC, including the control samples of CCH without extracts and the extracts of RBAC, YBAC, and PBAC for closed products and opened products.

Closed products showed intact L* during 10 days of storage, indicating the highest L* for the control samples of CHH and YBAC, whereas PBAC and RBAC exposed a similar L*. In turn, samples CCH + RBAC and CCH + PBAC showed an equal L*. Also, the extracts of RBAC, YBAC, and PBAC did not present color differences with the values of ΔE = 0, whereas the sample control CCH observed ΔE = 5. Therefore, betalain extracts on cottage cheese did not affect L* of the product.

Additionally, opened products exhibited a similar L* to closed products, showing no significative effects (*p* < 0.05); however, the color difference registered changes, observing the highest in the control extracts of YBAC, RBAC, and PBAC with ΔE = 20, whereas the change was lower in the samples CCH + PBAC and CCH + RBAC and CCH + YBAC, indicating ΔE = 5.

Consequently, the betalain extracts of RBAC, YBAC, and PBAC enhanced their permanency to oxygen and light exposure in cottage cheese because the presence of proteins in the cheese favored the color stability [19].

The application of betalains to cottage cheese has not been studied yet. However, here it showed that betalain extracts from RBAC, PBAC, and YBAC could be added to cottage cheese without altering the quality of product. In addition, the maintenance of betalains on cottage cheese is linked to interactions between betalains and proteins. Therefore, the preservation difference between the betalain extracts of RBAC, PBAC, and YBAC on cottage cheese describes a distinct retention capacity and retention mechanisms on this matrix. However, further studies could support this claim.

The mechanism of betalains’ interaction with proteins has been not established yet; however, the interaction of polysaccharides and milk proteins could describe the retention of betalains in proteins. The mechanism is based on the adsorption process and/or the interactions between molecules [38]. At nearly neutral pH levels, adsorption is predominant and favored by a competitive union between species or by limited thermodynamic compatibility between the protein and a polysaccharide. In turn, the interaction between molecules is described by the complexation of a polysaccharide with the adsorbed protein, hydrogen bonding, and electrostatic interactions between a polysaccharide and the protein at pH levels.

Previous reports of betalain application on dairy products also have showed the colorant capacity of betalains, registering color permanence and improvements in the properties in the products [1,2,3,4]; however, the reported studies are scarce, and distinct conditions and different tests are exposed in the published information. In this case, Ghasempour et al. [17] reported the positive effect of different formulations of betalains from red beetroot (RBE) and basil seed gum (BSG) in a probiotic yogurt. Authors found the best formulation of betalain application, containing 0.4% BSG and 0.1% RBE, to enhance the yogurt properties. The probiotic viability of yogurt increased (>60%) during the 21 days of storage time, whereas antioxidant activity, textural properties were maintained, and the viscosity of the product samples was enhanced by up to ~5000 cp.

Schneider-Teixeira et al. [18] carried out a colorimetric study on the application of the betacyanin extracts from Alternanthera brasiliana on yogurt. The authors observed the maintenance of betacyanin on yogurt, which was attributed to milk–polyphenol interactions. The extracted color was stable at the yogurt pH range, showing a berry-like color.

In turn, Güneşer [19] showed the stability of betalains from beetroots during the thermal treatment of cow milk, achieving moderate stability (70–140 min, at 70–80 °C).

Betalain extracts from Amarant and beetroot were also studied on ice-cream formulations, measuring the colorimetric characteristics of the product. Authors observed the color maintenance of betalains at up to 60 days of storage (−22 °C) of ice-cream, showing high a* values. In addition, positive effects were observed in the products [20].

The effectiveness of betalains as pigments on other food matrices was analyzed in Attia et al. [34]. The authors evaluated the incorporation of encapsulated red beetroot extract in jelly and ice sherbets. Color stability was found after 180 days of storage at −20 °C; in addition, the sensory study showed the acceptance of the product. Mehta et al. [22] analyzed the coloration of encapsulated betalains from prickly pear in gummies. The data were compared with commercial synthetic color treatments, resulting in pigmented products with similar colors without alteration of the product. Rodríguez-Sánchez et al. [39] incorporated betaxanthins from yellow pitaya (*S. pruinosus*) fruit as a coloring for drinks and jelly gummies, observing betaxanthin stability in stored products at low temperatures and under dark conditions. The betalains were more stable in the gummies due to low water activity. Currently, Chaari et al. [21] have developed an indicator film based on betacyanin from beetroot peel extract to monitor stored beef meat and to enhance the antioxidant characteristics of a product. The film provided real-time monitoring, showing coloration changes in correspondence with meat variations samples and microbial counts during the storage period.

## 4. Conclusions

The shelf-life of the betalain extracts of RBAC (red beetroot peel), YBAC (golden beetroot peel), and PBAC (purple prickly pear peel) were evaluated, measuring the capacity of their use as natural pigments of cottage cheese to obtain attractive and functional foods.

The extracts of RBAC resulted in the highest betalain content, predominating betacyanin, followed by YBAC with the high betaxanthin content and PBAC with the hightest betacyanin content.

The pH stability for the extracts was pH4–7; the thermal stability of the extracts of RBAC and PBAC was <90 °C, whereas YBAC exposed >125 °C. Under oxygen and light exposure, the extracts were stable at <10 days; however, YBAC exhibited low resistance under these conditions.

At 4 °C and amber storage, the shelf-life of RBAC showed betalain maintenance and a red-violet color at 87% at 180 days. YBAC was 95% with a yellow color, and PBAC maintained 60% of betalains with a violet color. The first order kinetics of extracts indicated the betalain degradation and color > 360 days; however, PBAC showed complete betalain degradation at this time.

The application of betalain extracts on cottage cheese exposed no changes in betalains and color in closed products at 10 days of storage at 4 °C. In regard to opened products, PBAC maintained the maximum amount of betalains (90%), followed by PBAC (75%); however, YBAC showed 60% in regard to betalain retention and color.

The obtained data contributed to different aspects of food areas, such as use of agro-industrial residues for betalain extraction. In addition, the extraction and conservation method of betalain extracts provided tools for betalain preparation.

Other information contributing to betalain knowledge includes their application on cottage cheese for food coloration and betalain stability improvements.

Also, betaxanthin production from golden beetroot and its capacity for application in foods is an attractive topic in the food area, providing knowledge which will be used in future research.

## Figures and Tables

**Figure 1 foods-14-00228-f001:**
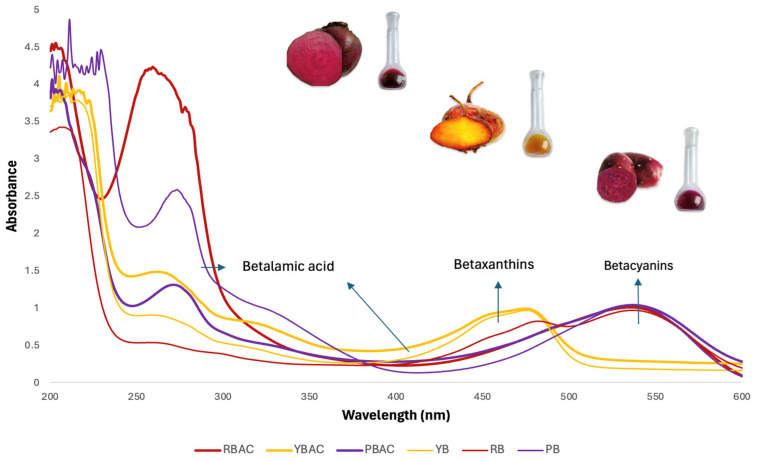
Uv-vis spectra and images of betalain extracts from red beetroot, golden beetroot, and purple prickly pear from samples of RBAC, YBAC and PBAC.

**Figure 2 foods-14-00228-f002:**
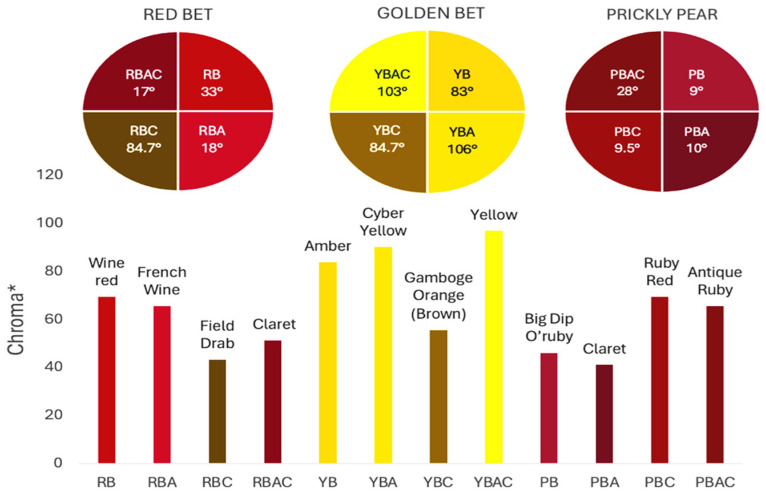
The color characteristics Chroma* and Hue° of the betalain extracts of RB, RBA, RBC RBAC, YB, YBA, YBC, YBAC, PB, PBA, PBC, and PBAC.

**Figure 3 foods-14-00228-f003:**
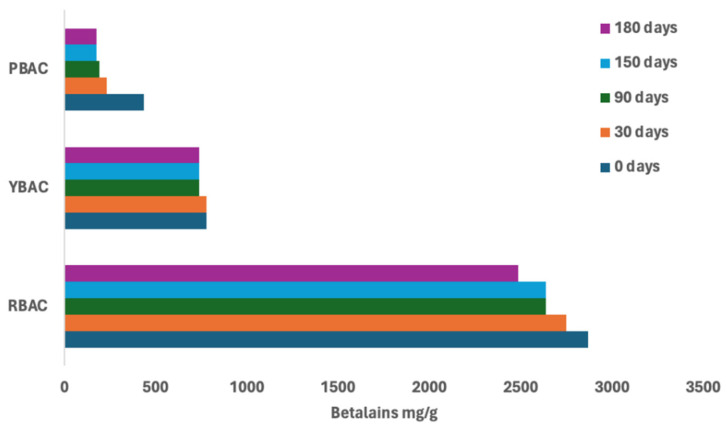
Kinetics of betalains maintenance in extracts RBAC, YBAC, and PBAC.

**Figure 4 foods-14-00228-f004:**
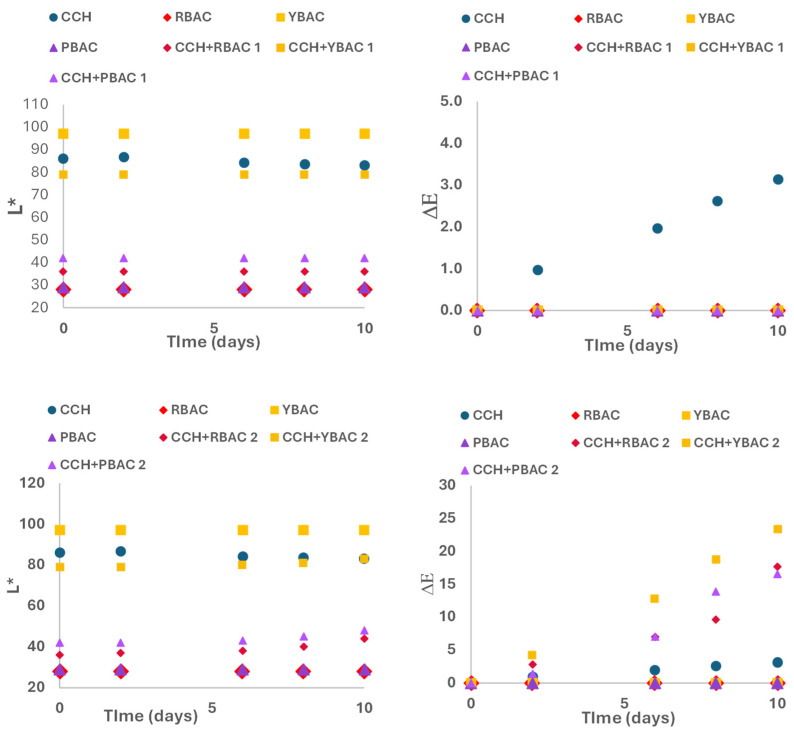
Luminosity L* and color difference ΔE from samples of cottage cheese with betalain extracts; conforming samples CCH + PBAC, CCH + RBAC and CCH-YBAC for closed products (above) and opened products (below).

**Table 1 foods-14-00228-t001:** The betalain content and physicochemical characteristics of betalain extracts from red beetroot, golden beetroot, and purple prickly pear.

Betalains Extract	Yield (mL/100 g FW)	Ph	Betaxanthins (mg/g)	Betacyanins (mg/g)	Total Betalains (mg/g)	°Bx	TDS (g/L)
RB	30	6.7 ± 0.01	392.3 ± 2.7	522.2 ± 6.2	914.5 ± 8.9	12 ± 0.1	94.7 ± 1
RBA	30	4.3 ± 0.01	666.73 ± 5.2	1068.07 ± 8.8	1734.8 ± 14	12 ± 0.5	94.6 ± 2
RBC	6	6.5 ± 0.02	267.11 ± 1	ND	267.1 ± 1	4 ± 1	44 ± 2
RBAC	6	4.3 ± 0.3	ND	2866.97 ± 5.04	2866.9 ± 5.04	61 ± 1	243.1 ± 4
YB	80	6.8 ± 0.01	86.4 ± 0.63	ND	86.4 ± 0.63	12 ± 0.1	15 ± 1
YBA	80	4.1 ± 0.1	105.94 ± 0.29	ND	105.9 ± 0.29	12 ± 0.1	34.2 ± 1
YBC	16	6.8 ± 0.1	ND	ND	ND	4 ± 0.1	7 ± 1
YBAC	16	4.5 ± 0.2	776.38 ± 0.35	ND	776.3 ± 0.35	61 ± 0.1	68.7 ± 1
PB	44	5.7 ± 0.5	ND	193.2 ± 13.42	193.2 ± 13.42	12 ± 0.5	111.6 ± 1
PBA	44	4.1 ± 0.3	ND	223.9 ± 13.24	223.9 ± 13.24	11.5 ± 1	185.5 ± 3
PBC	9	6.4 ± 0.5	ND	83.6 ± 0.43	83.6 ± 0.43	27.4 ± 1	189.2 ± 2
PBAC	9	4.3 ± 1	ND	433.9 ± 12	433.9 ± 12	10 ± 2	237.2 ± 4

**Table 2 foods-14-00228-t002:** The shelf-life of betalains in the extracts of red beetroot, yellow beetroot, and purple prickly pear extracts (RBAC, YBAC, and PBAC), as well as control extracts.

Source	Shelf-Life (Storage)	Betalains Content (mg/g)			
Red beetroot	Extracts	RBAC	RB	RBA	RBC
	t = 0 days	2867 ± 5	914 ± 9	1735 ± 14	267 ± 1
	t = 180 days	2485 ± 1	0	693.9 ± 14	0
	Betalains maintenance % after 180 days of storage	87	0	40	0
Yellow beetroot	Extracts	YBAC	YB	YBA	YBC
	t = 0 days	776 ± 1	86 ± 1	106 ± 1	0
	t = 180 days	791 ± 1	0	0	0
	Betalains maintenance % after 180 days of storage	95	0	0	0
Purple prickly pear	Extracts	PBAC	PB	PBA	PBC
	t = 0 days	434 ± 12	193 ± 13	224 ± 13	84 ± 0.5
	t = 180 days	177 ± 5	44 ± 2	78.5 ± 2	0
	Betalains maintenance % after 180 days of storage	50	23	35	0

**Table 3 foods-14-00228-t003:** Betalains retention and color difference in the extracts of RBAC, YBAC, and PBAC by pH, temperature, oxygen, and light exposure.

Betalain Extracts		RBAC			YBAC			PBAC		
Exposure Factors	Order	Betalains (mg/g)	Betalains Retention (%)	Color Difference ΔE	Betalains (mg/g)	Betalains Retention (%)	Color Difference ΔE	Betalains (mg/g)	Betalains Retention (%)	Color Difference ΔE
pH	4	2867 ± 0.5	100	0	776 ± 0.3	100	0	434 ± 0.3	100	0
	7	1442 ± 0.2	50	27	727 ± 0.5	94	10	420 ± 0.1	97	10
	8	1021 ± 0.1	36	60	696 ± 0.3	90	15	425 ± 0.6	60	49
	9	501 ± 0.1	18	72	672 ± 0.8	87	25	421 ± 0.4	47	57
Temperature (°C)	20	2801 ± 0.5	98	8	776 ± 0.3	100	0	115 ± 0.5	91	86
	90	2437 ± 0.6	85	38	776 ± 0.3	100	0	75 ± 0.3	85	90
	100	1581 ± 0.5	55	53	776 ± 0.3	100	0	14 ± 0.1	50	>100
	125	433 ± 0.5	20	74	776 ± 0.3	100	0	5 ± 0.1	1	>100
Oxigen and light (Days)	0	2867 ± 0.2	100	0	776 ± 0.3	100	0	434 ± 0.4	100	0
	10	2181 ± 1.3	76	35	384 ± 0.7	50	40	286 ± 1.6	66	26
	20	800 ± 1.6	250	75	297 ± 2.9	38	50	286 ± 2.1	56	32

**Table 4 foods-14-00228-t004:** Colorimetric stability of extracts RBAC, PBAC and YBAC under exposure to pH, temperature, oxygen and light.

	pH	T (°C)	Oxigen/Ligth (days)
RBAC		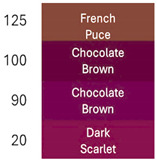	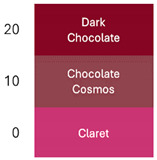
PBAC	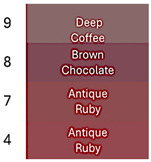	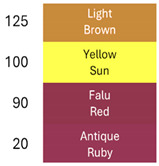	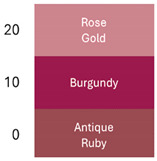
YBAC	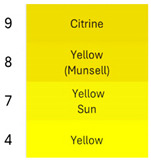	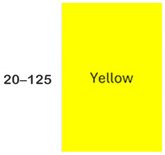	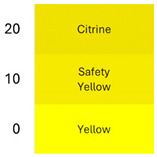

**Table 5 foods-14-00228-t005:** Shelf-life of the betalain extracts of RBAC, YBAC, and PBAC on cottage cheese.

Sample	Images of Pigmented Cotage Cheese		Total Betalains in Cotage Cheese (mg/g)		Color Maintanance in Cotage Cheese	
Time (days)	0	10	0	10	0	10
CCH + RBAC	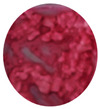	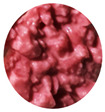	56 ± 0.03	52 ± 0.04		
CCH + YBAC	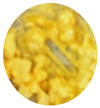	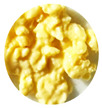	31 ± 0.39	19 ± 0.7		
CCH + PBAC	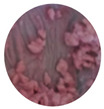	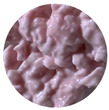	45 ± 0.39	38 ± 0.83		

## Data Availability

The original contributions presented in this study are included in the article. Further inquiries can be directed to the corresponding authors.

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
