# Peer review of "Conservation Analysis and Colorimetric Characterization of Betalain Extracts from the Peel of Red Beetroot, Golden Beetroot, and Prickly Pear Applied to Cottage Cheese"

_foods, 2025, doi:10.3390/foods14020228_

Round 1
Reviewer 1 Report
Comments and Suggestions for Authors
In the abstract, a sentence briefly including the conclusions is missing. Perhaps the materials and methods section could be further summarized to give more emphasis to both the results and the conclusions.
In the introduction, consider expanding on the issues associated with synthetic pigments to better highlight the importance of natural pigments.
In the materials and methods, sections as “preparation and conservation of extracts” and “physicochemical characteristics of the betalain extracts and extraction yield” could be divided for improved clarity.
In the line 341 could be replaced "the extracts were added with betalains to enhance stability" with "betalains were added to the extracts to enhance stability " for improved clarity.
Avoid redundancies; the sensitivity of betalains to pH and temperature, are repeated across multiple sections.
In the discussion section, consider expanding the comparison with similar studies on the use of betalains in dairy products.
In the discussion, consider explaining in more detail how factors like proteins in the cheese might have influenced color stability and why purple prickly pear betalains exhibited lower retention compared to beetroot betalains., citing relevant research.
Author Response
Manuscript ID foods-3393995
Comments for the author
REVIEWER 1
- In the abstract, a sentence briefly including the conclusions is missing. Perhaps the materials and methods section could be further summarized to give more emphasis to both the results and the conclusions.
RESPONSE: Thank you very much for the comments.
We structured a new version of manuscript, including
- In the introduction, consider expanding on the issues associated with synthetic pigments to better highlight the importance of natural pigments.
RESPONSE: Thank you very much for the comments. The new version of manuscript contains a paraph including synthetic pigments, Lines 47-49.
- In the materials and methods, sections as “preparation and conservation of extracts” and “physicochemical characteristics of the betalain extracts and extraction yield” could be divided for improved clarity.
RESPONSE: Thank you very much for the comments.
We divided in two sections the preparation and conservation of betalain extracts. Lines 116-126 and 128-148.
- In the line 341 could be replaced "the extracts were added with betalains to enhance stability" with "betalains were added to the extracts to enhance stability " for improved clarity.
RESPONSE: Thank you very much for the comments.
The paragraph indicated was replaced. Lines 350-351.
- Avoid redundancies; the sensitivity of betalains to pH and temperature, are repeated across multiple sections.
RESPONSE: Thank you very much for the comments.
We performed a review to remove redundancies in the section 3.4 correspondent to color stability.
- In the discussion section, consider expanding the comparison with similar studies on the use of betalains in dairy products.
RESPONSE: Thank you very much for the comments.
We included other studies on the use of betalains to compare de obtained information. Lines 572-597.
- In the discussion, consider explaining in more detail how factors like proteins in the cheese might have influenced color stability and why purple prickly pear betalains exhibited lower retention compared to beetroot betalains, citing relevant research.
RESPONSE: Thank you very much for the comments.
We included the paragraph lines 553-570 to explain the influence of cheese in the betalains stabilization.
Reviewer 2 Report
Comments and Suggestions for Authors
Manuscript titled “Conservation analysis and colorimetric characterization of betalain extracts from peel of red beetroot, golden beetroot, and prickly pear applied on cottage cheese” by López-Solórzano et al. presents an interesting and innovative study on the colorimetric characteristics and stability of betalain extracts derived from the peels of red beetroot, golden beetroot, and prickly pear. The approach of applying these extracts to cottage cheese to assess their ability to maintain betalains and pigmentation is a relevant and intriguing topic for the food industry. The potential of these natural colorant sources and the evaluation of their stability and resistance under storage conditions is a valuable contribution to the field and could be of interest to the journal’s readers. The study has good potential, but it requires significant revision.
In detail:
1) The keywords should not be identical to the title to improve searchability and ensure coverage of relevant topics. The current keywords are redundant, as they are exactly the same as the title. I would suggest using more specific terms that capture key aspects of the article.
2) The manuscript contains several typographical and writing errors, which make certain sections difficult to follow. These need to be corrected to improve the readability and professionalism of the article (ex. Lines 49, 63, 108, 116, 119, 120, 133… equation 2). Check the entire text.
3) Introduction, lines 64-72. The introduction would benefit from a more critical analysis of the existing literature. While the authors cite relevant studies, these works should be more critically evaluated to better highlight the novelty of the current study. A deeper comparison with similar studies in the field of natural colorants and their application in food products is recommended. This would help contextualize the current study within existing research and provide clearer justification for the work presented. There are several studies that were not cited, which could help strengthen the manuscript and improve its overall quality. For example: https://doi.org/10.1007/s11130-021-00915-6 .
4) Materials and methods. The article lacks specific details on the reagents and materials used. It is important to mention where the reagents and materials were obtained from. Information on the brands or suppliers should also be provided. This will enhance the reproducibility of the study and allow other researchers to replicate the experiment more easily (lines 104-105, 115…). Check the entire text.
5) Lines 197-199: “color properties and stability of extracts was measured by colorimetric properties L*, a*, b* to obtain L*, Hue and Chroma*…” Why L*? The sentence is not clear.
6) Line 202: What is “W”?
7) Results and Discussion (lines 232-242) section should not be presented as a simple numbered list but rather as a well-organized narrative that ties the data together and critically interprets the findings.
8) The presentation of Table 2 and Table 5 is not clear, and they need to be reformatted for better readability and comprehension. These tables should be reorganized to ensure that the data are presented in a more structured and consistent manner.
Author Response
RESPONSE TO REVIEW
Manuscript ID foods-3393995
Comments for the author
REVIEWER 2
Manuscript titled “Conservation analysis and colorimetric characterization of betalain extracts from peel of red beetroot, golden beetroot, and prickly pear applied on cottage cheese” by López-Solórzano et al. presents an interesting and innovative study on the colorimetric characteristics and stability of betalain extracts derived from the peels of red beetroot, golden beetroot, and prickly pear. The approach of applying these extracts to cottage cheese to assess their ability to maintain betalains and pigmentation is a relevant and intriguing topic for the food industry. The potential of these natural colorant sources and the evaluation of their stability and resistance under storage conditions is a valuable contribution to the field and could be of interest to the journal’s readers. The study has good potential, but it requires significant revision.
In detail:
- The keywords should not be identical to the title to improve searchability and ensure coverage of relevant topics. The current keywords are redundant, as they are exactly the same as the title. I would suggest using more specific terms that capture key aspects of the article.
RESPONSE: Thank you very much for the comments.
According to the suggestions, we replaced two key words.
- The manuscript contains several typographical and writing errors, which make certain sections difficult to follow. These need to be corrected to improve the readability and professionalism of the article (ex. Lines 49, 63, 108, 116, 119, 120, 133… equation 2). Check the entire text.
RESPONSE: Thank you very much for the comments.
We reviewed the entire paper to correct the indicated errors.
- Introduction, lines 64-72. The introduction would benefit from a more critical analysis of the existing literature. While the authors cite relevant studies, these works should be more critically evaluated to better highlight the novelty of the current study. A deeper comparison with similar studies in the field of natural colorants and their application in food products is recommended. This would help contextualize the current study within existing research and provide clearer justification for the work presented. There are several studies that were not cited, which could help strengthen the manuscript and improve its overall quality. For example: https://doi.org/10.1007/s11130-021-00915-6.
RESPONSE: Thank you very much for the comments.
Lines 78-93 from introduction were included in the manuscript to describe the novelty of the investigation. The suggested reference was included in line 493-496.
- Materials and methods. The article lacks specific details on the reagents and materials used. It is important to mention where the reagents and materials were obtained from. Information on the brands or suppliers should also be provided. This will enhance the reproducibility of the study and allow other researchers to replicate the experiment more easily (lines 104-105, 115…). Check the entire text.
RESPONSE: Thank you very much for the comments.
We check the text of indicated lines, to enhance the manuscript. However, the lines containing reagents were not modified because other regents were not required during the investigation.
- Lines 197-199: “color properties and stability of extracts was measured by colorimetric properties L*, a*, b* to obtain L*, Hue and Chroma*…” Why L*? The sentence is not clear.
RESPONSE: Thank you very much for the comments.
We corrected the indicated lines.
- Line 202: What is “W”?
RESPONSE: Thank you very much for the comments.
We replaced W by weight.
- Results and Discussion (lines 232-242) section should not be presented as a simple numbered list but rather as a well-organized narrative that ties the data together and critically interprets the findings.
RESPONSE: Thank you very much for the comments.
We were critic in the description of the data supporting the manuscript.
- The presentation of Table 2 and Table 5 is not clear, and they need to be reformatted for better readability and comprehension. These tables should be reorganized to ensure that the data are presented in a more structured and consistent manner.
RESPONSE: Thank you very much for the comments.
We placed some titles in the rows of the tables to enhance their description